# Eremophilane-Type Sesquiterpenes from a Marine-Derived Fungus Penicillium Copticola with Antitumor and Neuroprotective Activities [note 1]

**DOI:** 10.3390/md20110712

**Published:** 2022-11-13

**Authors:** Jianping Zhang, Dong Liu, Aili Fan, Jian Huang, Wenhan Lin

**Affiliations:** 1State Key Laboratory of Natural and Biomimetic Drugs, Peking University, Beijing 100191, China; 2Institute of Ocean Research, Ningbo Institute of Marine Medicine, Peking University, Ningbo 315010, China

**Keywords:** marine fungus, *Penicillium copticola*, copteremophilanes A–J, structure elucidation, antitumor activity, neuroprotection

## Abstract

Chemical examination of a marine sponge-associated *Penicillium copticola* fungus resulted in the isolation of ten undescribed eremophilanes, namely copteremophilanes A–J (**1**–**10**), along with two new glycosides, 5-glycopenostatin F (**11**) and 5-glucopenostatin I (**12**). Their structures were determined by extensive spectroscopic data, in association with ECD data and chemical conversions for configurational assignments. Analogs **1**, **2**, and **10** represent a group of uncommon skeletons of eremophilanes with an aromatic ring and a methyl migration from C-5 to C-9, and analogs **11** and **12** are characteristic of a PKS scaffold bearing a glucose unit. The incorporation of a chlorinated phenylacetic unit in **3**–**9** is rarely found in nature. Analog **7** showed neuroprotective effect, whereas **8** exhibited selective inhibition against human non-small cell lung cancer cells (A549). This study enriched the chemical diversity of eremophilanes and extended their bioactivities to neuroprotection.

## 1. Introduction

Farnesyl diphosphate (FPP) biogenetically generates sesquiterpenes with diverse scaffolds via various ring rearrangements. Eremophilanes are one class of sesquiterpenes characterized by the presence of a bicyclic backbone with irregular rule to assemble FPP in association with a methyl migration from C-10 to C-5 [1,2]. Eremophilane analogues extensively distribute in nature. Apart from the eremophilanes derived by plants of which the genus *Ligularia* is dominated to produce eremophilanes [3], terrestrial- or marine-originated fungi emerge as the additional sources to produce relevant analogs. It is noteworthy that most fungal eremophilanes are enantiomeric to the corresponding entities from plants [1], and marine-associated fungi have the potential to generate structurally unique analogs [4,5,6]. Eremophilanes possess wide range of bioactivities, such as phytotoxins, antimicrobials, protein inhibitors, immunomodulators and cytotoxins. In marine-derived fungi, eremophilanes with chemical diversity are widely distributed in fungal genera of *Acremonium* [6], *Penicillium* [7,8,9], *Cochliobolus* [10], *Phomopsis* [11], and *Cryptosphaeria* [12]. These findings imply marine-derived fungi as a potential source to generate structurally unique and bioactive eremophilanes. With the aim to continue our discovery of bioactive natural products from marine-associated organisms, a marine sponge (*Xestospongia testudinaria*)-associated fungus strain *Penicillium copticola* WZXY-m122-9 was selected for chemical examination. The LC-MS/MS data of the EtOAc extract of the cultured fungus were processed into a molecular network using MZmine and the GNPS platform (http://gnps.ucsd.edu, accessed on 20 June 2022), which allowed the formation of the spectral nodes into clusters (Appendix A). Annotation of the nodes in a cluster with *m*/*z* values of 200 to 300 by GNPS MS/MS spectral library matched eremophilanes, including dehydropetasol (*m*/*z* 233 [M + H]^+^), dihydrosporogen AO-1 (*m*/*z* 251 [M + H]^+^), hydroxyphomenone (*m*/*z* 267 [M + H]^+^), sporogen AO-1 (*m*/*z* 248 [M + H]^+^) and penicilleremophilane X (*m*/*z* 279 [M + H]^+^). In addition, a cluster with the nodes ranging from *m/z* 430 to 480 presenting chlorine feature ([M]^+^/[M + 2]^+^ = 3:1) did not hit in the database, suggests a group of untapped metabolites. The scaled-up fermentation and extensive chromatographic separation of the EtOAc extract resulted in the isolation of ten new eremophilane-type sesquiterpenes, along with two new PKS glucosides (Figure 1). Herein, we report the structural determination of the new compounds (Appendix A) and the bioactivities of antitumor cell lines and neuroprotection.

## 2. Results

### 2.1. Structure Elucidation of New Compounds

Copteremophilane A (**1**) has a molecular formula of C_14_H_18_O_2_ as determined by the HRESIMS and NMR data, requiring six degrees of unsaturation. The ^13^C NMR and DEPT data afforded a total of 14 carbon resonances, including six aromatic carbons for a phenyl group, a ketone, as well as seven alkyl carbons, of which three were classified into methyl groups. A *tetra*-substituted aromatic ring was recognized by the presence of two *meta*-coupled aromatic protons, H-6 (*δ*_H_ 7.61, brs) and H-8 (*δ*_H_ 7.55, brs). The substitution of a methyl group at C-9 (*δ*_C_ 136.4) and an acetyl group at C-7 (*δ*_C_ 140.8) was confirmed by the HMBC correlations from H_3_-14 (*δ*_H_ 2.24, s) to C-8 (*δ*_C_ 126.8), C-9, and C-10 (*δ*_C_ 134.8), and both H-6 and H-8 to carbonyl carbon C-11 (*δ*_C_ 198.3), along with H_3_-12 (*δ*_H_ 2.53, s) to C-7 (*δ*_C_ 140.8) and C-11. The COSY correlations established a segment in ring A from C-1 to C-4, and this unit was fused to C-5 (*δ*_C_ 141.1) and C-10 positions of the aromatic ring to form a cyclohexene ring due to the HMBC correlations of C-5 and C-10 to both H_2_-1 (*δ*_H_ 2.62, 2.69) and H-4 (*δ*_H_ 2.78, dq, *J* = 2.6, 7.2 Hz). In addition, a methyl group at C-4 and a hydroxy group at C-3 were confirmed by the COSY correlations from H-4 to H-3 (*δ*_H_ 3.66, dt, *J* =2.6, 4.9 Hz) and H_3_-15 (*δ*_H_ 1.23, d, *J* = 7.2 Hz), along with the HMBC correlations from H_3_-15 to C-3 (*δ*_C_ 69.8), C-4 (*δ*_C_ 41.2), and C-5. The NOE correlation between H_3_-15 and H-3 was indicative of a *trans*-orientation of H_3_-15 toward OH-3. Based on the modified Mosher method, the (*R*)-MPA and (*S*)-MPA esters of **1** were synthesized. Calculation of the *Δδ* (*δ*_R_ − *δ*_S_) values resulted in 3*R* configuration (Figure 2). Thus, a 4*R* configuration was suggested with the help of the NOE data. The similar data of the experimental ECD compared to that calculated for (3*R*, 4*R*)-**1** (Figure 3A) further supported the configurational assignment.

The molecular formula (C_15_H_20_O_2_) of copteremophilane B (**2**) was afforded by the HRESIMS and NMR data. The NMR data of both **1** and **2** (Table 1) were comparable, except for the distinction of the substituent at C-7 (*δ*_C_ 136.3). A hydroxyisopropene unit was identified by the NMR resonances for two olefinic carbons and a hydroxymethyl group, in association with the HMBC correlations from the hydroxymethyl protons H_2_-12 at *δ*_H_ 4.28 (s) to C-11 (*δ*_C_ 148.3) and C-13 (*δ*_C_ 110.1) and between H_2_-13 (*δ*_H_ 5.22, 5.34) and C-12 (*δ*_C_ 63.0). The location of the hydroxyisopropene unit at C-7 was confirmed by the HMBC correlations of H-6 (*δ*_H_ 7.06, brs) and H-8 (*δ*_H_ 7.03, brs) to C-11 and from H_2_-13 to C-7. The same relative configuration of both **1** and **2** was evident from the NOE correlations between H-3 and H_3_-15. The similar ECD data suggested the same absolute configuration for both **1** and **2**. This was supported by the calculated ECD data (Figure 3B), in which the experimental ECD data were consistent with those calculated for (3*R*, 4*R*)-**2**.

Copteremophilane C (**3**) was determined to have a molecular formula of C_23_H_25_ClO_7_ by the HRESIMS and NMR data, requiring 11 degrees of unsaturation. Diagnostic 2D NMR data revealed **3** to be assembled by two moieties. Based on the 2D NMR data, one of them was identified as a phomenone unit, a co-isolated eremophilane-type sesquiterpene which was structurally characterized by the presence of an α,β-unsaturated ketone with an epoxy group at ring B [13]. The second moiety contained six aromatic carbons for a phenyl unit, a methylene and a carbonyl carbon, along with two *meta*-coupling aromatic protons H-2′ (*δ*_H_ 6.70, d, *J* = 2.1 Hz) and H-6′ (*δ*_H_ 6.67, d, *J* = 2.1 Hz) and the methylene protons H_2_-7′ (*δ*_H_ 3.50, brs). The HMBC correlations from H_2_-7′ to a carbonyl carbon at C-8′ (*δ*_C_ 171.2), C-2′ (*δ*_C_ 121.0) and C-6′ (*δ*_C_ 115.5) and from H-2′ and H-6′ to the aromatic carbons established a tri-substituted phenylacetic segment. Additional HMBC correlations from two phenol protons OH-4′ (*δ*_H_ 9.01, brs) and OH-5′ (*δ*_H_ 9.76, brs) to the aromatic carbons, respectively, in association with the molecular composition, allowed the assignment of a 3-chloro-4,5-dihydroxyphenylacetic moiety. The linkage of the acyl unit to C-3 for an ester formation was evident from the HMBC correlation between H-3 (*δ*_H_ 4.82, dt, *J* = 4.4, 11.2 Hz) and C-8′. The same relative configuration of **3** as phomenone was identified by the NOE relationships of both H_3_-14 (*δ*_H_ 1.22, s) and H_3_-15 (*δ*_H_ 0.90, d, *J* = 6.8 Hz) with H-3 and H-6 (*δ*_H_ 3.40, s) and between H-6 and H_2_-13 (*δ*_H_ 5.08, 5.20) (Figure 4). Comparison of the experimental and calculated ECD data (Figure 3C) suggested **3** possessing *R* configuration for C-3, C-4, C-5, C-6, and C-7.

The molecular composition of copteremophilane D (**4**) showed the same as that of **3** according to the HRESIMS data. The NMR data of both compounds (Table 2 and Table 3) were very similar, and the 2D NMR data established the gross structure of **4** also containing two segments which were identical to those of **3**. The distinction was attributed to the 3-chloro-4,5-dihydroxyphenylacetic moiety location. The chemical shifts of C-3 (*δ*_C_ 69.0 vs. *δ*_C_ 73.2 of **3**) and C-12 (*δ*_C_ 64.9 vs. *δ*_C_ 62.1 of **3**) in addition to the HMBC correlation between H_2_-12 (*δ*_H_ 4.70, 4.71) and a carbonyl carbon C-8′ (*δ*_C_ 171.1) allowed the location of the acyl group at C-12. The similar NOE data of both **3** and **4**, along with the comparable ECD data of **4** to those calculated for (3*R*, 4*R*, 5*R*, 6*R*, 7*R*)-**4** (Figure 3D), assigned the same configuration of both **3** and **4**.

The molecular formula of copteremophilane E (**5**) was established as C_23_H_25_ClO_6_ from the HRESIMS (*m*/*z* 431.1259 [M − H]^−^, calcd 431.1261) data, containing 11 degrees of unsaturation. The NMR data of **5** (Table 2 and Table 3) were comparable to those of **4**, and the NMR resonances for phomenone moiety of both compounds were identical. The acyl moiety was identified as 3-chloro-4-hydroxyphenylacetic group based on the presence of an ABX spin system for the aromatic protons at H-3′ (*δ*_H_ 6.88, d, *J* = 8.2 Hz), H-2′ (*δ*_H_ 6.97, dd, *J* = 1.5, 8.2 Hz), and H-6′ (*δ*_H_ 7.18, d, *J* = 1.5 Hz), and the HMBC correlations from the methylene protons H_2_-7′ (*δ*_H_ 3.54, s) to C-1′ (*δ*_C_ 126.1), C-2′ (*δ*_C_ 131.0), C-6′ (*δ*_C_ 129.4) and the carbonyl carbon C-8′ (*δ*_C_ 171.2), along with the correlations of H-2′ to the nonprotonated carbons C-3′ (*δ*_C_ 119.8) and C-4′ (*δ*_C_ 152.5). The similar NOE and ECD data (Figure 3E) suggested the same configuration for both **4** and **5**.

The structure of copteremophilane F (**6**) was determined to be a 5′-methoxylated analog of **4** due to the comparable NMR data of both analogs, with the exception of an additional methoxy group in **6**. The methoxy protons at *δ*_H_ 3.78 (s) showed HMBC correlation to C-5′ (*δ*_C_ 148.4) and the NOE relationship with H-6′ (*δ*_H_ 6.77, d, *J* = 1.9 Hz) confirmed a 3-chloro-4-hydroxy-5-methoxyphenylacetic unit, which was linked to C-12 (*δ*_C_ 64.5) on the basis of the HMBC correlation between H_2_-12 (*δ*_H_ 4.70, 4.76) and the carbonyl carbon C-8′ (*δ*_C_ 170.6). The similar NOE data of **6** as those of **4** and **5** in association with the comparable ECD data of **6** to those calculated for (3*R*, 4*R*, 5*R*, 6*R*, 7*R*)-**6** identified the same configuration of **6** as **4**.

Copteremophilane G (**7**) has a molecular formula of C_23_H_27_ClO_6_ as determined by the HRESIMS data. The NMR data demonstrated **7** possessing two moieties. Analyses of 2D NMR data (Appendix A) revealed the sesquiterpene unit being identical to JBIR-27 [9], while the second unit was consistent with a 3-chloro-4,5-dihydroxyphenylacetic group. The linkage of the acyl unit to nucleus moiety at C-14 (*δ*_C_ 66.6) was deduced by the HMBC correlation between H_2_-14 (*δ*_H_ 4.19, 4.43) and the carbonyl carbon C-8′ (*δ*_C_ 171.2). The NOE relationships from H_2_-14 to H_3_-15 (*δ*_H_ 0.98, d, *J* = 6.9 Hz) and H-7 (*δ*_H_ 3.19, dd, *J* = 4.0, 11.0 Hz) and between H_3_-15 and H-3 (*δ*_H_ 3.49, m) (Figure 4) suggested the same relative configuration of the nucleus moiety as JBIR-27 [9]. The comparable ECD data of **7** to those calculated for (3*R*, 4*R*, 5*S*, 7*S*)-**7** (Figure 3G) agreed the absolute configuration of **7** as *R* for C-3 and C-4 and *S* for C-5 and C-7.

The molecular formula of copteremophilane H (**8**) was the same as **7** due to the same HRESIMS data, requiring ten degrees of unsaturation. The NMR data of **8** were comparable to those of **7**. The distinction was found in ring B, where two methyl groups were recognized for H_3_-12 (*δ*_H_ 1.80, d, *J* = 1.5 Hz) and H_3_-13 (*δ*_H_ 2.02, d, *J* = 1.5 Hz). The HMBC correlations of H_3_-12/H_3_-13 to C-7 (*δ*_C_ 127.3) and C-11 (*δ*_C_ 142.9) indicated **8** to be a stereoisomer of **7** with an olefinic transformation from C-11/C-13 to C-7/C-11. The NOE correlations from H_3_-15 (*δ*_H_ 1.06, d, *J* = 6.8 Hz) to H_2_-14 (*δ*_H_ 4.15, s) and H-3 (*δ*_H_ 3.46, td, *J* = 5.4, 10.6 Hz) clarified the same relative configuration of **8** and **7**. Comparison of the experimental ECD data to those calculated for (3*R*, 4*R*, 5*S*)-**8** (Figure 3H) established 3*R*, 4*R* and 5*S* configurations for **8**.

The molecular formula of copteremophilane I (**9**) was determined as C_23_H_29_ClO_6_ by the HRESIMS data, requiring nine degrees of unsaturation. The NMR data of **9** resembled those of **3** but distinguished by ring B of sesquiterpene unit. Hydroxylations at C-7 (*δ*_C_ 73.7) and C-8 (*δ*_C_ 68.2) were clarified by the COSY coupling between H-8 (*δ*_H_ 3.60, brd, *J* = 4.5 Hz) and H-9 (*δ*_H_ 5.40, brd, *J* = 4.5 Hz) together with the HMBC correlations from H_2_-6 (*δ*_H_ 1.68, 1.69) to C-5 (*δ*_C_ 38.6), C-7, C-8, C-11 (*δ*_C_ 150.9) and C-14 (*δ*_C_ 20.2). The NOE correlations from H-3 (*δ*_H_ 4.73, m) to H_3_-14 (*δ*_H_ 1.08, s) and H_3_-15 (*δ*_H_ 0.79, d, *J* = 6.8 Hz) and from H_3_-12 (*δ*_H_ 1.76, s) to H_3_-14 and H-8 suggested the same face of both hydroxy groups. The absolute configuration of **9** was assigned by the similar ECD data of **9**, compared to those calculated for (3*R*, 4*R*, 5*R*, 7*S*, 8*S*)-**9**.

Copteremophilane J (**10**) was determined to have a molecular formula of C_23_H_27_ClO_6_ by the HRESIMS data, containing ten degrees of unsaturation. The NMR data featured a homolog of **3**, and the 2D NMR data assigned a 3-chloro-4,5-dihydroxyphenylacetic group at C-3 as the case of **3**. In regard to the sesquiterpene moiety, a phenyl ring was assigned to ring B due to the presence of six aromatic carbons as well as the HMBC correlations of the aromatic protons H-6 (*δ*_H_ 7.12, brs) and H-8 (*δ*_H_ 7.06, brs) to the aromatic carbons. The HMBC correlations of methyl protons H_3_-14 to C-8, C-9, and C-10 in association with the NOE correlation between H_2_-1 and H_3_-14 substituted a methyl group at C-9. The remaining NMR resonances were attributed to a 1,2-dihydroxypropane unit, which was deduced by the HMBC correlations from H_2_-12 (*δ*_H_ 3.37, s) to C-11 (*δ*_C_ 73.9) and C-13 (*δ*_C_ 22.6). The location of this unit at C-7 (*δ*_C_ 145.0) was confirmed by additional HMBC correlations from H_3_-13 (*δ*_H_ 1.36, s) and H_2_-12 to C-7. The NOE relationships between H-3 (*δ*_H_ 4.87, ddd, *J* = 2.3, 4.4, 6.3 Hz) and H_3_-15 (*δ*_H_ 1.21, d, *J* = 7.2 Hz) suggested a *cis*-orientation of H-3 toward H_3_-15. The absolute configuration of C-12 was determined as *S* on the basis of the Mo(AcO)_4_ induced ECD data (ICD) (Figure 3J), in which a positive Cotton effect at 330 nm was observed.

Compound **11** has a molecular formula of C_28_H_42_O_8_ as established by the HRESIMS data. The DEPT ^13^C NMR spectrum exhibited a total of 28 carbon resonances, involving 6 olefinic carbons, a ketone carbon, 7 sp^3^ oxygenated carbons, and 14 other alkyl carbons. Analyses of the 2D NMR data established a nucleus structure which was identical to penostatin F [13]. This assignment was evident from the COSY correlations along with the HMBC correlations to establish a 4-methylcycloocta-3,5-dien-1-one (Figure 5). Additional COSY relationship between H-7 and H-8 in association with the HMBC correlations from H-7 to C-2 and C-3 and from H-2 to C-1 and C-9 fused a cyclohexene across C-1 and C-8. A spin system from H_2_-4 to H-7 via H-5 and H_2_-6 along with the HMBC correlation between H-5 and C-3 clarified a cyclopentane ring fused to C-3 and C-7, and C-5 (*δ*_C_ 78.7) was oxygenated. The location of an n-heptane unit at C-14 was based on additional COSY and HMBC data. The remaining six oxygenated sp3 carbons were attributed to a sugar unit, which was identified as glucose according to the comparison of the NMR data with those of authentic sample. It was linked to C-5 based on the HMBC correlation between H-5 (*δ*_H_ 4.38, t, *J* = 5.0 Hz) and C-1′ (*δ*_C_ 102.1). The NOE correlations between H-5 and H-8 (*δ*_H_ 2.86, t, *J* = 5.6 Hz) and between H-7 (*δ*_H_ 3.11, m) and H-14 (Figure 6) suggested the relative configuration of the nucleus moiety to be identical to penostatin F. The *J*_H-1′/H-2′_ value (7.9 Hz) suggested a β-form of Glc unit. Acidic hydrolysis of **11** derived an aglycone and a sugar, the former was identical to penostatin F based on the comparable NMR data and specific rotation and the latter was identical to authentic glucose due to the TLC and HPLC chromatographic comparison.

Compound **12** has the same molecular formula as **11** according to the HRESIMS data. The NMR data of **12** (Table 4) were almost superimposed to those of **11**, suggesting the structural similarity of both compounds. Analysis of 2D NMR data established the same planar structure of **11** and **12**. The similar NOE data with the exception of the correlation between H-5 and H-7 instead of between H-7 and H-1′ of **11** suggested the stereoisomers of both compounds (Figure 6). However, the opposite rotation ([α]^20^_D_ −10 for **11** and [α]^20^_D_ +12 for **12**) and the opposite Cotton effects (Figure 7) suggested an enantiomeric form with the exception of the chiral center at C-5, implying the nucleus part of **12** to be consistent with penostatin I [13].

In addition, the phenylacetic acids, including sporogen AO-1 [14], phomenone [15], JBIR28 [9], JBIR27 [9], 3-chloro-4,5-dihydroxyphenylacetic acid, 3-chloro-4-hydroxyphenylacetic acid, and 3-chloro-4-hydroxy-5-methoxyphenylacetic acid [16], were isolated and identified on the basis of spectroscopic data.

### 2.2. Bioassays

Fungus-derived eremophilanes gain potential activities against cancer cell lines for specific aspects in cancer therapies [17,18]. To evaluate the relevant effects of the isolated compounds, an MTT method was performed to detect the cytotoxic activities of compounds toward human non-small cell lung cancer cells (A549), human colon cancer cells (HCT-8) and human breast cancer cells (MCF-7). As shown in Table 5, eremophilane analogs possessing specific selection to inhibit tumor cell lines were observed. Copteremophilane H (**8**) with an acyl unit at C-14 selectively inhibited A549 cells with IC_50_ value of 3.23 μM. However, analog **7** with an olefinic rearrangement from C-7/C-11 of **8** to C-11/C-13 totally attenuated the inhibitory effect. Analogs **4** and **5** with a phenylacetic unit at C-12 of phomenone showed selective inhibition against HCT-8 cell line. It is noteworthy that **4** with a 3-chloro-4,5-dihydroxy-phenylacetic unit exhibited higher activity than **5** which possesses a 3-chloro-4-hydroxyphenylacetic unit, whereas **6** with a 5-methoxy-4-hydroxy-3-chlorophenylacetic unit was inactive toward the three tumor cell lines. These findings suggested that the substituents at the phenylacetic unit are sensitive to affect the activities. In contrast with the bioactive analogs, the eremophilane nuclei and the phenylacetic units alone had weak to no antitumor effects, implying the necessity for the combination of eremophilane moiety with phenylacetic unit to improve antitumor effects. In addition, the irregular eremophilanes **1**, **2** and **10** with an aromatic ring B attenuated the inhibitory activities.

In previous work, we even reported eremophilanes possessing the inhibitory effects against LPS-induced NO production in macrophages [19]. Oxidative stress has been implicated in the pathology of Alzheimer’s disease (AD), and accumulation of β-amyloid (Aβ) causes oxidative stress [20,21]. Aβ is considered as a major pathological factor to induce Alzheimer’s disease [22,23]. To evaluate the neuroprotective effects of analogs, we extend the bioassay of analogs to A*β*_25-35_-stimulated PC12 cell line. PC12 cells are the pheochromocytoma-derived cell line which maintains a differentiated neuroendocrine phenotype [24]. Firstly, the A*β*_25-35_-induced PC12 cell injury was detected by the CCK8 assay. Aβ_25-35_ induced cell death in a dose-dependent manner, and around 50% inhibition of PC12 cells was observed after the treatment of Aβ_25-35_ (40 μM) for 24 hrs. Thus, Aβ_25-35_ (40 μM) was selected as an optimal dose for subsequent experiments. Alternatively, eremophilanes (50 μM) alone maintained normal PC12 cell morphology and cell numbers, suggesting low cytotoxicity. Most analogs showed inhibitory activities against Aβ_25-35_-induced cell death, while **7** is the most active to increase the viability of Aβ_25-35_-induced PC12 cells (Table 6). Lactate dehydrogenase (LDH) as a stable enzyme in the cytosol is quickly released into the medium upon damage of plasma membrane, and malondialdehyde (MDA) as an essential product of lipid peroxidation tends to elevate free radical-mediated myocardial cell injury [25,26]. Thus, low levels of LDH and MDA can protect PC12 cells against oxidative stress. In the Aβ_25-35_ induced PC12 cell model, LDH release was significantly increased compared to the control group. Analog **7** significantly reduced LDH release dose-dependently. Meanwhile, the intracellular level of MDA, a marker of lipid peroxidation, was also reduced (Figure 8). 

## 3. Discussion

Biogenetically, petasol is considered as a precursor to derive relevant analogs. Dehydrogenation of petasol generates 6-dehydropetasol as an intermediate, which follows olefinic rearrangement and methyl migration and subsequent oxidation to afford the aromatic analogs **1**, **2** and **10**. Epoxidation 6-dehydropetasol derives sporogen-OA1, which was isolated from this fungus. 12-Hydroxylation of sporogen-OA1 followed by various acylation generates **3**–**6**. By the similar manner, 14-hydroxylation of petasol and then acylation derives **7**, which follows olefinic migration affords **8** (Figure 1).

## 4. Materials and Methods

### 4.1. General Experimental Procedures 

Optical rotations were recorded on an Autopol-III automatic polarimeter (Rudolph Research Co., Ltd., Hackettstown, NJ, USA). UV spectra were recorded on a Cary 300 spectrometer. IR spectra were measured on a Thermo Nicolet Nexus 470 FT-IR spectrometer. ECD spectra were measured on a JASCO J-815 spectropolarimeter. ^1^H and ^13^C NMR spectra together with 2D NMR spectra were measured on Bruker Advance NMR spectrometers (400 MHz or 600 MHz for ^1^H and 100 MHz or 125 MHz for ^13^C, respectively). Chemical shifts are expressed in *δ* referenced to the solvent peaks of DMSO-*d*_6_ (^1^H at *δ*_H_ 2.50 and for ^13^C at *δ*_C_ 39.5). HRESIMS spectra were obtained on a Bruker APEX IV 70 eV FT-MS spectrometer and on a Waters Xevo G2 Q-TOF spectrometer fitted with an ESI source (acquisition range:100–1000, acquisition: start time 0 and end time 4, voltages: ESI^+^ 2 kV and ESI^−^ 1.5–2 kV, external standard: HCOONa). DAD HPLC was performed on Waters e2695 separations module with Waters 2998 photodiode array detector and Thermo BDS column (250 × 4.6 mm, 5 μm). Column chromatography (CC) was performed using ODS (50 μm, Daiso), silica gel (200–300 mesh, Qingdao Marine Chemistry Co., Ltd., Qingdao, China), and Sephadex LH-20 (Amersham Pharmacia Biotech AB, Staffanstorp, Sweden). Precoated silica gel plates (Qingdao Marine Chemistry Co., Ltd.) were used for TLC analyses. Solvents used for isolation are all analytical grade. Semipreparative HPLC was performed on an Alltech 426 pump using a Uvis-201 detector, and the Prevail C_18_ column (semipreparative, 5 μm) was used for separation. 

### 4.2. Fungal Strain and Identification

Fungal strain WZXY-m122-9 was isolated from the sponge of *Xestospongia testudinaria*, which was collected from Weizhou island in May 2016. The fungal identification was based on the ITS gene sequence which showed 100% similarity to a *Penicillium copticola* clone using Blast. The ITS gene sequence was deposited in GenBank. The strain is preserved at the State Key Laboratory of Natural and Biomimetic Drugs, Peking University.

### 4.3. Fermentation and Extraction

The fungal strain WZXY-m122-9 was cultured on flasks (120 × 500 mL) in rice medium with 45 g rice and 40 mL of 3.3% sea-salt in each flask at 25 °C for 4 days to obtain fresh mycelia and spores. They were then inoculated in 250 mL Erlenmeyer flasks (×10) containing 50 mL PDB medium to obtain seed medium after culture in a rotary shaker set to 120 rpm at 25 °C for 3 days. The seed culture was inoculated in 600 × 250 mL Erlenmeyer flasks, each containing 30 g rice and 30 mL of distilled artificial seawater. After 16 days under static conditions at 25 °C, the fermented culture was extracted with EtOAc (2 × 250 mL) twice. The organic solvent was evaporated to obtain an extract (20.0 g), which was suspended in 10% H_2_O of MeOH, and then extracted with cyclohexane to remove lipids. The MeOH solution was evaporated under reduced pressure to obtain an extract (10.0 g). This extract was dissolved in H_2_O and then extracted by EtOAc to yield EtOAc extract (4.8 g) after concentrated under vacuum. The EtOAc extract was fractionated upon silica gel column (3 × 25 cm) eluting with Petroether-EtOAc (3:1) to afford three fractions: FA, FB (0.5 g) and FC (1.2 g). Fraction FA (2.4 g) was fractionated upon C_18_ (ODS) column, eluting with MeOH-H_2_O (1:3) to yield four subfractions (FA1 to FA4). FA2 (670 mg) was repeatedly separated by semi-preparative HPLC (YMC-packed C_18_, 5 μm, 250 × 10 mm, 2 mL/min, UV detection at 210 nm) with MeCN-H_2_O (1:4, *v*/*v*) as a mobile phase to yield sporogen AO-1 (125 mg, R_t_ = 27.5 min), phomenone (18.0 mg, R_t_ = 25.5 min), JBIR28 (4.0 mg, R_t_ = 23.6 min), and JBIR27 (13.5 mg, R_t_ = 29.4 min). Subfraction FA2 (195 mg) was separated by semi-preparative HPLC (YMC-packed C_18_, 5 μm, 250 × 10 mm, 2 mL/min, UV detection at 210 nm) eluting with MeCN-H_2_O (1:3, *v*/*v*) to obtain **1** (10.5 mg, R_t_ = 24.8 min), **2** (5.5 mg, R_t_ = 29.1 min), **11** (6.9 mg, R_t_ = 28.5 min), **12** (5.5 mg, R_t_ = 28.7 min), **3** (9.2 mg, R_t_ = 30.2 min), **4** (5.8 mg, R_t_ = 31.1 min), and **5** (10.5 mg, R_t_ = 30.4 min). Subfraction FA3 (55 mg) was purified by a semi-preparative HPLC (YMC-packed C_18_, 5 μm, 250 × 10 mm, 2 mL/min, UV detection at 210 nm) eluting with MeOH-H_2_O (45:55, *v*/*v*) to afford **6** (7.9 mg, R_t_ = 22.4 min), **7** (9.2 mg, R_t_ = 23.6 min), **8** (6.2 mg, R_t_ = 27.0 min), **9** (7.5 mg, R_t_ = 25.5 min), and **10** (4.1 mg, R_t_ = 26.0 min). FA1 (140 mg) was separated upon C_18_ ODS column (C_18_, 10 μm, 2.5 × 30 cm), eluting 3-chloro-4,5-dihydroxyphenylacetic acid (7.6 mg), 3-chloro-4-hydroxyphenylacetic acid (5.5 mg), and 3-chloro-4-hydroxy-5- methoxyphenylacetic acid (3.1 mg).

Copteremophilane A (**1**): yellow amorphous; [*α*]^20^_D_ +62 (*c* = 0.2, MeOH); IR (KBr) v_max_ 3384, 2933, 2880, 1677, 1601, 1453, 1383, 1209 cm^−1^; ^1^H and ^13^C NMR data, see Table 1; HRESIMS *m*/*z* 219.1380 [M + H]^+^ (calcd for C_14_H_19_O_2_, 219.1385). 

(*R*)-MPA ester of **1**: white powder; ^1^H NMR (400 MHz, DMSO-*d*_6_) *δ* 2.61 (2H, m, H-1), 1.94–2.08 (2H, m, H-2), 4.93 (1H, m, H-3), 2.85 (1H, m, H-4), 7.50 (1H, s, H-6), 7.58 (1H, s, H-8), 3.28 (3H, s, H-12), 2.25 (3H, s, H-13), 1.11 (1H, d, *J* = 6.5 Hz, H-14).

(*S*)-MPA ester of **1**: white powder; ^1^H NMR (400 MHz, DMSO-*d*_6_) *δ* 2.48 (2H, m, H-1), 1.75–2.92 (2H, m, H-2), 4.99 (1H, m, H-3), 3.02 (1H, m, H-4), 7.58 (1H, s, H-6), 7.61 (1H, s, H-8), 3.28 (3H, s, H-12), 2.14 (3H, s, H-13), 1.23 (1H, d, *J* = 6.5 Hz, H-14).

Copteremophilane B (**2**): yellow amorphous; [*α*]^20^_D_ +26 (*c* = 0.2, MeOH); IR (KBr) v_max_ 3396, 2931, 1685, 1604, 1453, 1384, 1208 cm^−1^; ^1^H and ^13^C NMR data, see Table 1; HRESIMS *m*/*z* 233.1543 [M + H]^+^ (calcd for C_15_H_21_O_2_, 233.1542).

Copteremophilane C (**3**): yellow amorphous; [*α*]^20^_D_ +90 (*c* = 0.3, MeOH); IR (KBr) v_max_ 3283, 2929, 1731, 1673, 1596, 1499, 1436, 1208, 1204 cm^−1^; ^1^H and ^13^C NMR data, see Table 2 and Table 3; HRESIMS *m*/*z* 449.1369 [M + H]^+^ (calcd for C_23_H_26_ClO_7_, 449.1367).

Copteremophilane D (**4**): yellow amorphous; [*α*]^20^_D_ +28 (*c* = 0.2, MeOH); IR (KBr) v_max_ 3260, 2941, 1732, 1670, 1596, 1500, 1436, 1205 cm^−1^; ^1^H and ^13^C NMR data, see Table 2 and Table 3; HRESIMS *m*/*z* 447.1208 [M − H]^−^ (calcd for C_23_H_24_ClO_7_, 447.1211).

Copteremophilane E (**5**): yellow amorphous; [*α*]^20^_D_ +28 (*c* = 0.2, MeOH); IR (KBr) v_max_ 3291, 2927, 1735, 1669, 1596, 1510, 1425, 1384, 1153 cm^−1^; ^1^H and ^13^C NMR data, Table 2 and Table 3; HRESIMS *m*/*z* 431.1259 [M − H]^−^ (calcd for C_23_H_24_ClO_6_, 431.1261).

Copteremophilane F (**6**): yellow amorphous; [*α*]^20^_D_ +46 (*c* = 0.2, MeOH); IR (KBr) v_max_ 3315, 2926, 1735, 1669, 1505, 1384, 1285, 1145 cm^−1^; ^1^H and ^13^C NMR data, see Table 2 and Table 3; HRESIMS *m*/*z* 461.1363 [M − H]^−^ (calcd for C_24_H_26_ClO_7_, 461.1367).

Copteremophilane G (**7**): yellow amorphous; [*α*]^20^_D_ +60 (*c* = 0.2, MeOH); IR (KBr) v_max_ 3268, 2939, 1725, 1674, 1499, 1436, 1291, 1206 cm^−1^; ^1^H and ^13^C NMR data, see Table 2 and Table 3; HRESIMS *m*/*z* 433.1415 [M − H]^−^ (calcd for C_23_H_26_ClO_6_, 433.1418).

Copteremophilane H (**8**): yellow amorphous; [*α*]^20^_D_ +102 (*c* = 0.2, MeOH); IR (KBr) v_max_ 3256, 2928, 1685, 1436, 1383, 1297, 1209 cm^−1^; ^1^H and ^13^C NMR data, see Table 2 and Table 3; HRESIMS *m*/*z* 433.1421 [M − H]^−^ (calcd for C_23_H_28_ClO_6_, 433.1418).

Copteremophilane I (**9**): yellow amorphous; [*α*]^20^_D_ +54 (*c* = 0.2, MeOH); IR v_max_ (KBr) cm^−1^: 3386, 2928, 1720, 1436, 1384, 1292, 1206, 1150, 1024; ^1^H and ^13^C NMR data, see Table 2 and Table 3; HRESIMS *m*/*z* 435.1578 [M − H]^−^ (calcd for C_23_H_28_ClO_6_, 435.1574).

Copteremophilane J (**10**): yellow amorphous; [*α*]^20^_D_ +54 (*c* = 0.2, MeOH); IR (KBr) v_max_ 3432, 2996, 2913, 1661, 1436, 1406, 1312 cm^−1^; ^1^H and ^13^C NMR data, see Table 1; HRESIMS *m*/*z* 457.1395 [M + Na]^+^ (calcd for C_23_H_27_ClO_6_Na, 457.1390). 

5-Glycopenostatin F (**11**): yellow amorphous; [*α*]^20^_D_ +12 (*c* 0.2, MeOH); IR v_max_ (KBr) cm^−1^: 3402, 2926, 2855, 1716, 1674, 1384, 1205; ^1^H and ^13^C NMR data, see Table 4; HRESIMS *m*/*z* 507.2958 [M + H]^+^ (calcd for C_28_H_43_O_8_, 507.2958).

5-Glycopenostatin I (**12**): yellow amorphous; [*α*]^20^_D_ -10 (*c* 0.2, MeOH); IR v_max_ (KBr) cm^−1^: 3372, 2926, 1716, 1456, 1383, 1341, 1218; ^1^H and ^13^C NMR data, see Table 4; HRESIMS *m*/*z* 507.2958 [M + H]^+^ (calcd for C_28_H_43_O_8_, 507.2958).

### 4.4. Hydrolysis of 11 and 12

Compound **11** (1.5 mg) was dissolved in HCl (2 M, 10 mL) to stir at 100 °C for 4 h monitored by TLC. After reaction time, the acidic solution was extracted by 5 mL EtOAc. EtOAc was concentrated under vacuum to yield 0.7 mg hydrolyzed product (**11a**), and the acidic solution was freeze-dried and solubilized in MeOH for further analysis. Compound **12** (1.3 mg) was hydrolyzed by the same protocol as for **11**.

### 4.5. Mosher Reaction 

Compound was dissolved in anhydrous CHCl_3_, and then *R*-MPA (equal mol), DMAP and DCC were added to react at room temperature for 12 hrs. The product was purified by silica gel column eluting with petroleum ether-EtOAc (2:1) to yield *R*-MPA ester. *S*-MPA ester of compound was synthesized by the same protocol as for *R*-MPA ester. Calculation of the chemical shift difference (Δ*δ* = *δ*_R_ − *δ*_S_) allowed the assignment of configuration of the stereogenic center of secondary alcohol [27].

### 4.6. Snatzke Method 

Compound was weighted to two portions (0.2 mg for each). One portion was dissolved in 0.5 mL DMSO for the detection of ECD curve at wavelength 250–500 nm, and the second portion was dissolved in 0.5 mL DMSO with 0.5 mg/mL Mo(OAc)_4_ for the detection of the ECD curve within 30 min [28].

### 4.7. Cytotoxic Detection

Cell viability was evaluated using the MTT assay according to the manufacturer’s instruction. In brief, cells were plated at a density of 1.0 × 10^4^ cells per well in 96-well plates and allowed to attach overnight. Then, the cells were treated with or without compound at the indicated concentration. After 48 h, MTT solution containing 1 mg/mL and MTT 100 μL was added into each well and incubated for 2 h at 37 °C. Then, the medium was changed with the same volume of DMSO. After incubation, absorbance is read at 570 nm for MTT by a spectrophotometer and the quantity of formazan product is directly proportional to the number of living cells in culture.

### 4.8. Cell Viability Assay 

Rat pheochromocytoma (PC12) cells were cultured in Dulbecco’s modified Eagle medium (DMEM), and supplemented with 10% FBS, 100 U/mL penicillin, and 100 μg/mL streptomycin in a humidified atmosphere at 37 °C with 5% CO_2_. The cells were passaged every 3 days. PC12 cells (1 × 10^4^) were cultured for 24 h, and then the medium was replaced with serum-free DMEM medium. The PC12 cells were divided into three groups. In the model group, PC12 cells were treated using Aβ_25–35_ with different concentrations. Then, the cells were incubated for 12 h, 24 h and 48 h until there was a cell viability of up to 50%. PC12 cells without pretreatment were set up as control. The third group of PC12 was treated with different concentrations of compounds to detect the cell viability by CCK8 method. The model groups (Aβ_25–35_ induced PC12 cells) were treated with compounds in different concentrations, and the viability of PC12 cells was measured using cell counting kit-8 (Dojindo, Japan), according to the manufacturer’s instructions. Then, the medium was aspirated and cells in each group were incubated with 10 μL of CCK8 at 37 °C for 2 h. Afterwards, the absorbance was determined at a wavelength of 450 nm with a microplate reader (Thermo, MuLTiSKAN MK3, Waltham, MA USA).

### 4.9. Measurement of Malondialdehyde (MDA)

Cultured PC12 cells were initially seeded in 6-well plates for 24 h. The cells were then pre-incubated with or without compound, followed by incubation with Aβ_25-35_ (40 μM) for 24 h. The cultures were washed with ice-cold PBS and homogenized. The homogenate was centrifuged at 4 °C. The protein concentration in each sample was determined by the MDA Protein Assay Kit as a reference standard. The levels of MDA were determined according to the manufacturer’s instructions. Concentrations were normalized to the protein concentration expressed as a percentage of control samples.

### 4.10. Lactate Dehydrogenase (LDH) Release Assay

Cell injury was assessed through measuring the LDH activity in the supernatant of PC12 cells using an LDH kit according to the manufacturer’s protocol. In brief, double-distilled H_2_O, 0.2 μM pyruvic acid, matrix buffer and coenzyme I buffer were added in sequence at 48 h after the treatment by compound. The supernatant was collected after incubation for 30 min at room temperature. The absorbance at 450 nm was then measured with a microplate reader.

### 4.11. Statistical Analysis

Data are expressed as mean ± standard deviation (SD). Biostatistical analyses were conducted with SPSS 16.0 software. Statistical differences among groups were assessed by one-way analysis of variance (ANOVA). Differences were considered to be statistically significant at *p* < 0.05.

## 5. Conclusions

In summary, this work reports a chemical examination of the marine sponge-derived *P. copticola* fungus to afford 12 undescribed natural products, of which 10 are identified as eremophilanes and 2 are determined as glucosides. Eremophilanes **1**, **2** and **10** are characteristic of the eremophilanes with a phenyl moiety in ring B accompanying a methyl migration to C-9, which are uncommonly found in nature. Eremophilanes **3**–**9** feature the esterification of a chlorinated phenylacetic unit in backbone, and this is the second isolation of relevant analogs from fungi. Analogs **11** and **12** are characteristic of an unprecedented PKS scaffold bearing a glucose unit. The bioassay results revealed the inhibitory effects of the isolated eremophilanes against tumor cell lines related to the structure variation. Analog **8** only inhibited human non-small cell lung cancer cells (A549), and analogs **4** and **5** selectively inhibited HCT-8. The eremophilane nuclei or acyl moieties showed inactivity towards tumor cell lines. These findings suggest the incorporation of chlorinated phenylacetic units significantly enhances the antitumor effects of eremophilanes, and the substituted positions and the acyl compositions are sensitive for the selective inhibition. In addition, the noncytotoxic analogs, such as **7**, showed a neuroprotective effect. This study implies that eremophilanes have potential for development as antitumor or neuroprotective agents after structure modification.

## Figures and Tables

**Figure 1 marinedrugs-20-00712-f001:**
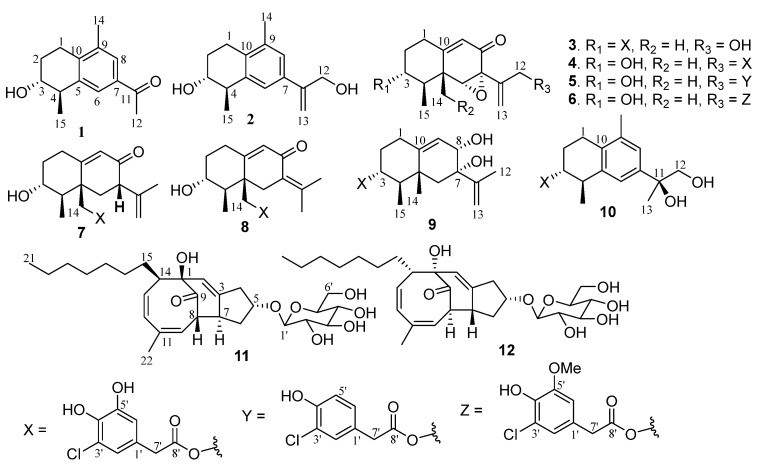
Structures of new compounds.

**Figure 2 marinedrugs-20-00712-f002:**
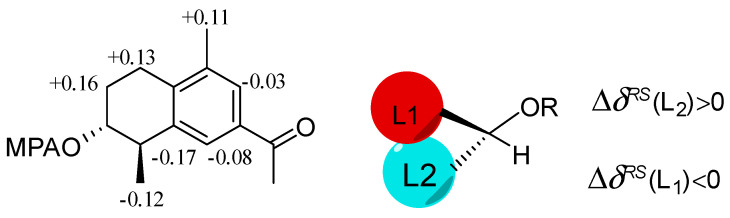
Δ*δ* (*δ*_R_ − *δ*_S_) values of MPA esters of **1**.

**Figure 3 marinedrugs-20-00712-f003:**
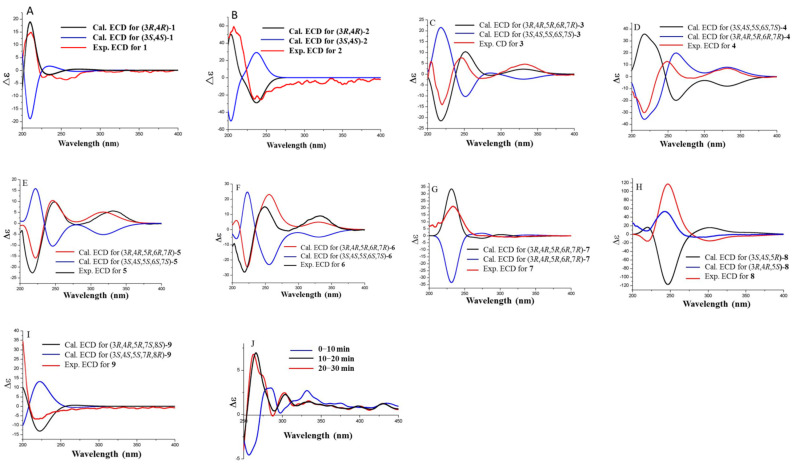
Experimental and calculated ECD spectra of **1**–**9** (**A**–**I**) and ICD spectra of **10** (**J**).

**Figure 4 marinedrugs-20-00712-f004:**
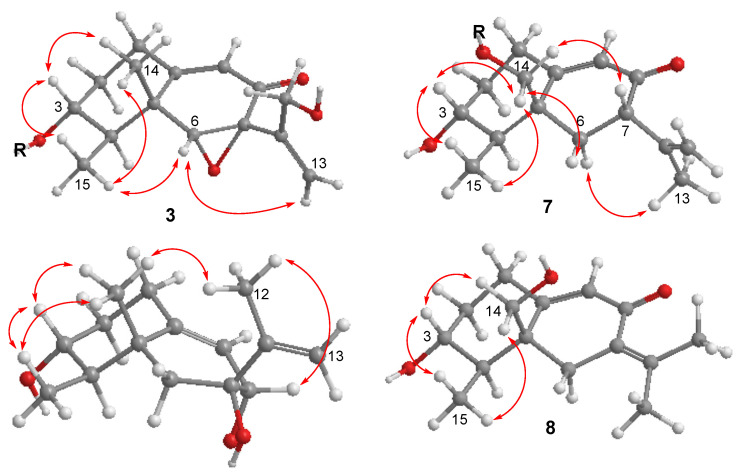
Key NOE correlations of **3** and **7**–**9**.

**Figure 5 marinedrugs-20-00712-f005:**
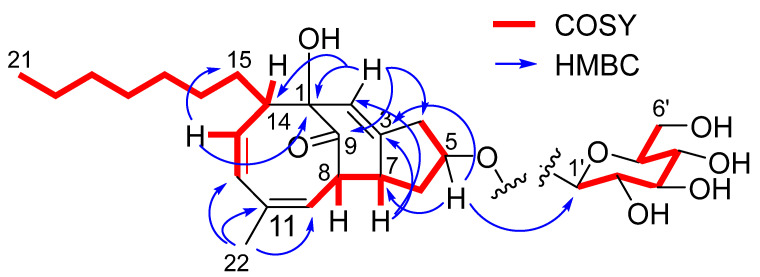
Key ^1^H-^1^H COSY and HMBC correlations of **11** and **12**.

**Figure 6 marinedrugs-20-00712-f006:**
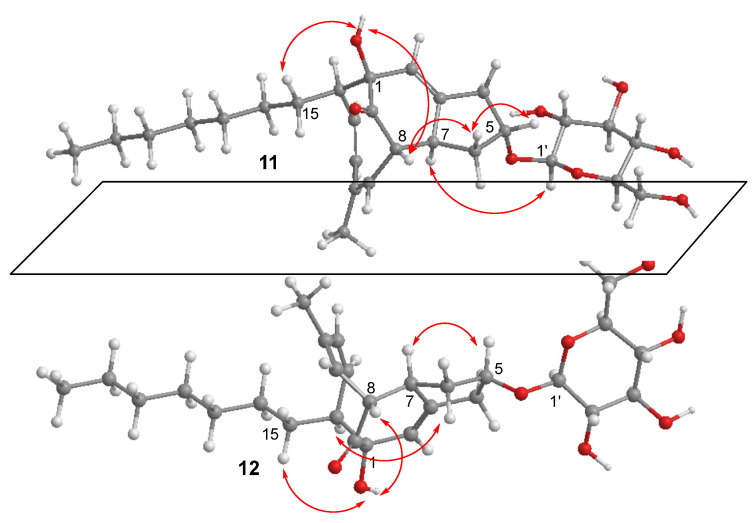
Key NOE correlations of **11** and **12**.

**Figure 7 marinedrugs-20-00712-f007:**
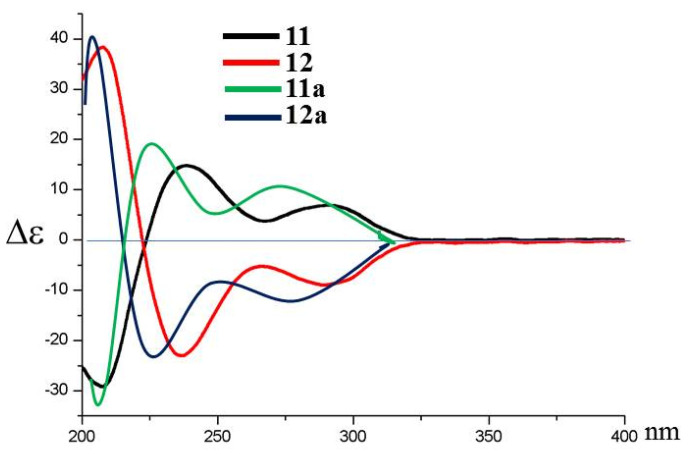
ECD spectra of **11** and **12** and their hydrolyzed products **11a** and **12a**.

**Figure 8 marinedrugs-20-00712-f008:**
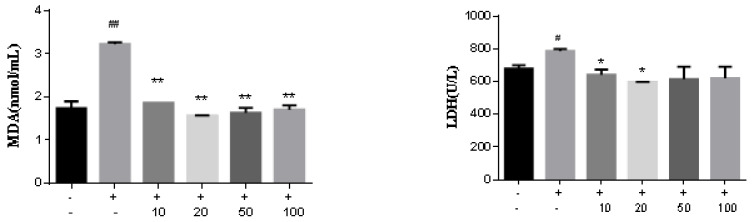
Analog **7** reduced the expression of MDA and LDH in Aβ_25-35_-induced PC12 cells. Statistical significance values are indicated as *, # *p* < 0.05, **, ## *p* < 0.01.

**Scheme 1 marinedrugs-20-00712-sch001:**
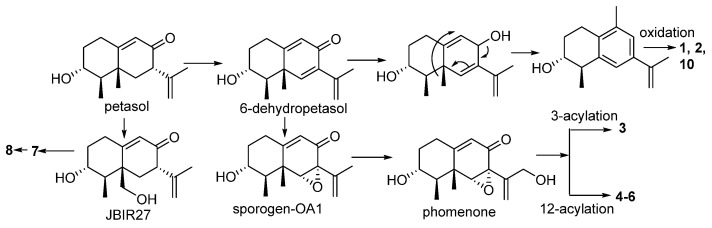
Biogenetic relationships of the isolated eremophilanes.

**Table 1 marinedrugs-20-00712-t001:** ^1^H and ^13^C NMR data of **1**, **2** and **10** in DMSO-*d*_6_.

	1		2		10	
	*δ* _H_	*δ* _C_	*δ* _H_	*δ* _C_	*δ* _H_	*δ* _C_
1	2.62, dt (4.0, 12.0) 2.69, td (6.0, 12.0)	23.7	2.50, dt (3.5, 12.0) 2.63, td (5.0, 12.0)	23.3	2.55, dt (4.0, 12.0) 2.62, td (4.0, 12.0)	22.4
2	1.80, m; 1.95, m	27.1	1.71, m; 1.90, m	27.7	1.90, m; 1.98, m	23.9
3	3.66, dt (2.6, 4.9)	69.8	3.60, dt (3.0, 5.0)	70.3	4.87, dt (4.4, 6.3)	74.6
4	2.78, dq (2.6, 7.2)	41.2	2.67, dq (3.0, 7.2)	41.2	2.91, dq (4.4, 7.2)	38.1
5		141.1		140.5		138.0
6	7.61, brs	127.1	7.06, brs	124.7	7.12, s	123.8
7		140.8		136.3		145.0
8	7.55, brs	126.8	7.03, brs	124.2	7.06, s	125.1
9		136.4		135.6		134.9
10		134.8		134.1		131.4
11		198.3		148.3		73.9
12	2.53, s	27.1	4.28, s	63.0	3.35, dd (4.0, 12.0) 3.37, dd (4.0, 12.0)	71.0
13			5.22, d (1.7) 5.34, d (1.7)	110.1	1.36, s	26.6
14	2.24, s	19.8	2.16, s	21.3	2.17, s	20.1
15	1.23, d (7.2)	21.4	1.12, d (7.2)	19.9	1.21, d (7.2)	21.7
1′						125.51
2′					6.66, d (1.9)	121.0
3′						120.3
4′						141.1
5′						146.9
6′					6.63, d (1.9)	115.6
7′					3.45, s	40.1
8′						171.4
OH-12					4.59, t (4.0)	

**Table 2 marinedrugs-20-00712-t002:** ^1^H NMR data of **3**-**9** in DMSO-*d*_6_.

	3	4	5	6	7	8	9
1	2.63, dt (2.0, 14.5) 2.36, td (2.0, 14.5)	2.28, td (3.0, 12.0) 2.54, dt (3.0, 12.0)	2.26, td (3.2, 14.1) 2.51, dt (3.2, 14.1)	2.25, td (3.4, 11.2) 2.50, dt (3.4, 11.2)	2.30, m 2.33, m	2.30, td (3.4, 12.0) 2.39 dt (3.4, 11.2)	2.07, m 2.29, m
2	1.36, m 2.04, m	1.25, m 1.99, m	1.19, m 1.98, m	1.20, m 1.98, m	1.31, m 2.00, m	1.29, m 2.01, m	1.27, m 1.97, m
3	4.82, dt (4.4, 11.2)	3.44, dt (5.0, 12.0)	3.42, dt (5.0, 10.5)	3.43, dt (5.0, 10.5)	3.49, m	3.46, td (5.4, 10.6)	4.73, m
4	1.82, dq (6.8,11.2)	1.53, dq (6.8, 12.0)	1.54, dq (6.8, 10.5)	1.54, dq (6.8, 10.5)	1.29, dq (6.9, 10.5)	1.37, dq (6.8, 10.6)	1.44, dq (6.8, 11.0)
6	3.40, s	3.38, s	3.38, s	3.41, s	1.80, t (11.0) 2.20, dd (4.0, 11.0)	2.24, d (15.0) 3.08, d (15.0)	1.68, d (12.6) 1.69, d (12.6)
7					3.19, dd (4.0, 11.0)		
8							3.60, brd (4.5)
9	5.75, d (1.7)	5.71, d (1.7)	5.70, d (1.7)	5.68, d (1.7)	5.78, d (1.3)	5.82, d (1.3)	5.40, d (4.5)
12	4.05, dd (4.0, 12.0) 4.12, dd (4.0, 12.0)	4.70, d (12.0) 4.71, d (12.0)	4.67, d (12.0) 4.74, d (12.0)	4.70, d (12.0) 4.76, d (12.0)	1.60, s	1.80, d (1.5)	1.76, s
13	5.08, d (1.5) 5.20, d (1.5)	5.29, d (0.9) 5.36, d (0.9)	5.28, d (1.1) 5.36, d (1.1)	5.38, d (1.1) 5.29, d (1.1)	4.68, brs 4.83, brs	2.02, d (1.5)	4.76, d (1.0) 4.85, d (1.0)
14	1.22, s	1.12, s	1.12, s	1.11, s	4.19, d (11.5) 4.43, d (11.5)	4.15, s	1.08, s
15	0.90, d (6.8)	1.11, d (6.8)	1.11, d (6.8)	1.11, d (6.8)	0.98, br d (6.9)	1.06, d (6.8)	0.79, d (6.8)
2′	6.70, d (2.1)	6.64, d (2.0)	6.97, dd (1.5, 8,2)	6.79, d (1.9)	6.61, d (1.6)	6.61, d (1.9)	6.68, d (1.9)
3′			6.88, d (8.2)				
6′	6.67, d (2.1)	6.62, d (2.0)	7.18, d (1.5)	6.77, d (1.9)	6.58, d (1.6)	6.57, d (1.9)	6.65, d (1.9)
7′	3.50, s	3.47, s	3.54, s	3.55, s	3.45, s	3.40, s	3.47, s
OH-3		4.70, br					
OH-12	4.83 t (4.0)						
OH-4′	9.01, brs	9.02, s	10.0, brs		9.00, s	8.99, s	
OH-5′	9.76, brs	9.74, s			9.80, s	9.69, s	
MeO				3.78, s			

**Table 3 marinedrugs-20-00712-t003:** ^13^C NMR data of **3**-**9** in DMSO-*d*_6_.

	3	4	5	6	7	8	9
1	29.9	30.9	30.9	30.3	31.4	31.2	30.1
2	31.7	35.8	35.8	35.4	35.7	35.8	32.9
3	73.2	69.0	69.2	68.4	69.5	69.6	74.7
4	41.9	44.8	44.8	44.3	50.7	49.3	47.9
5	41.2	41.2	41.2	40.7	43.1	44.8	38.6
6	68.3	69.2	68.9	68.6	39.1	38.3	40.6
7	61.7	61.2	61.2	60.7	50.7	127.3	73.7
8	192.2	192.1	192.2	191.6	198.0	190.2	68.2
9	120.7	120.0	120.0	119.4	126.4	128.6	121.2
10	163.7	166.2	166.6	165.6	164.3	163.1	143.0
11	145.3	139.3	139.3	138.8	144.0	142.9	150.9
12	62.1	64.9	64.8	64.5	20.3	22.6	19.4
13	111.9	117.1	117.2	116.8	114.3	23.0	110.4
14	18.1	18.4	18.4	17.9	66.6	66.6	20.2
15	11.3	11.7	11.8	11.2	11.4	11.9	11.0
1′	126.0	125.6	126.1	125.1	125.5	125.5	126.2
2′	121.0	120.3	131.0	122.1	120.9	121.0	121.0
3′	120.3	121.0	119.8	119.5	120.4	120.4	120.3
4′	141.2	141.3	152.5	141.6	141.3	141.2	141.1
5′	146.9	146.9	116.9	148.4	146.9	146.9	146.9
6′	115.5	115.6	129.4	111.8	115.5	115.6	115.4
7′	39.9	39.6	39.2	39.7	39.0	40.0	40.0
8′	171.2	171.1	171.2	170.6	171.2	171.2	171.3
OMe				56.6			

**Table 4 marinedrugs-20-00712-t004:** ^1^H and ^13^C NMR data of 11 and 12 in DMSO-*d*_6_.

No		11		12
	*δ* _C_	*δ* _H_	*δ* _C_	*δ* _H_
1	82.1		82.3	
2	124.9	5.50, br d (2.6)	125.3	5.50, br d (2.6)
3	146.1		144.9	
4	37.9	2.46, m 2.60, m	37.7	2.27, m 2.68, m
5	78.7	4.38, t (5.0)	78.1	4.34, m
6	39.6	1.44, m 2.43, m	39.5	1.45, m 2.57, m
7	48.6	3.11, m	48.5	2.79, m
8	50.4	2.86, t (5.6)	50.4	2.91, m
9	211.2		210.6	
10	128.3	5.58, dd (0.9, 6.4)	128.2	5.61, dd (0.9, 6.4)
11	129.4		129.5	
12	130.2	5.67, d (11.5)	130.3	5.68, d (11.5)
13	133.9	5.60, dd (9.1, 11.5)	133.9	5.34, dd (9.2, 11.5)
14	43.3	2.57, m	42.7	2.53, m
15	28.4	1.55, m	28.2	1.53, m
16	28.0	1.09, m 1.22, m	27.8	1.08, m 1.21, m
17	29.6	1.22, m	29.2	1.20, m
18	29.1	1.22, m	29.0	1.20, m
19	31.7	1.22, m	31.7	1.21, m
20	22.5	1.25, m	22.5	1.25, m
21	14.4	0.87, t (7.1)	14.4	0.85, t (7.1)
22	25.6	1.75, s	25.6	1.74, s
1’	102.1	4.22, d (7.9)	102.5	4.20, d (7.7)
2’	73.8	2.93, m	73.8	2.89, m
3’	77.2	3.16, m	77.3	3.13, m
4’	70.6	3.05, m	70.5	3.03, m
5’	77.4	3.08, m	77.4	3.03, m 3.08, m
6’	61.6	3.44, dt (5.7, 11.8) 3.67, dd (2.0, 11.8)	61.6	3.43, m 3.67, dd (2.0, 11.8)

**Table 5 marinedrugs-20-00712-t005:** Inhibitory effects of **1**-**10** against tumor cell lines.

		IC_50_ (μM)	
Comps	A549	HCT-8	MCF-7
**1**	>10	>10	>10
**2**	>10	>10	>10
**3**	>10	>10	>10
**4**	>10	5.4 ± 0.1	>10
**5**	>10	7.3 ± 0.1	>10
**6**	>10	>10	>10
**7**	>10	>10	>10
**8**	3.2 ± 0.1	>10	>10
**9**	>10	>10	>10
**10**	>10	>10	>10
taxol	0.2 ± 0.1	0.7 ± 0.3	0.2 ± 0.1

**Table 6 marinedrugs-20-00712-t006:** Aβ_25-35_-induced PC12 cell viability treated by analogs.

	% PC12 Cell Viability to Control					
μM	0	5.0	10.0	20.0	30.0	40.0
Aβ_25-35_	100	92.5	81.0	74.6	64.8	49.5
Aβ_25-35_ + **3**	50.0	53.7	55.8	62.1	63.7	65.5
Aβ_25-35_ + **4**	50.0	55.2	52.4	67.8	72.2	75.5
Aβ_25-35_ + **5**	50.0	50.0	50.0	50.0	50.0	50.0
Aβ_25-35_ + **6**	50.0	51.0	53.0	55.0	57.0	59.2
Aβ_25-35_ + **7**	50.0	56.2	60.5	74.6	78.6	84.3
Aβ_25-35_ + **8**	50.0	52.6	55.5	58.4	62.7	65.5
Aβ_25-35_ + **9**	50.0	50.0	50.0	50.0	50.0	50.0

## Data Availability

The authors confirm that the data supporting the findings of this study are available within the article and its Appendix A.

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
