# Peer review of "Eremophilane-Type Sesquiterpenes from a Marine-Derived Fungus Penicillium Copticola with Antitumor and Neuroprotective Activities†"

_marinedrugs, 2022, doi:10.3390/md20110712_

Round 1

Reviewer 1 Report

The manuscript entitled “Eremophilane-type sesquiterpenes from a marine-derived Penicillium copticola fungus with antitumor and neuroprotective activities” described the discovery of twelve new compounds from a marine fungus Penicillium copticola. The structures including their configurations were determined by the analysis of extensive spectroscopic data and/or ECD calculation, chemical conversions. Some of the analogs showed pretty unique structural features. Compounds 7 and 8 exhibited neuroprotective effect and inhibition against human non-small cell lung cancer cells (A549). Overall, this is a very comprehensive study, and the paper was properly written. Thus, I suggest that this manuscript may be suitable for publication in Marine Drugs after minor revision.

Here are some comments about the manuscript, which should be addressed by the authors.

Line 2, better revise into “a marine-derived fungus Penicillium copticola

Line 17, “unprecedented is a strong word, please revise this”

Line 19, better introduce the activity of 7 and then 8.

Line 44, please provide details about the in-house database comparison. NMR signals? MS or MSMS patterns?

Some figures are blurry, please provide figures with higher resolution. (Figures 4, 7, 8)

Line 90, double check Figure 3, panel C.

Author Response

Answer to referee-1

  • Line 2, better revise into “a marine-derived fungus Penicillium copticola

Response: Thank for the comment, I agree it, and the title in revised to “Eremophilane-type sesquiterpenes from a marine-derived fungus Penicillium copticola with antitumor and neuroprotective activities”.

  • Line 17, “unprecedented is a strong word, please revise this”

Response: Thank for the comment. I deleted the word “unprecedented” although the scaffold is rarely found from nature.

  • Line 19, better introduce the activity of 7 and then 8.

Response: I did the revision of the sentence as “Analog 7 showed neuroprotective effect, whereas 8 exhibited selective inhibition against human non-small cell lung cancer cells (A549)”.

  • Line 44, please provide details about the in-house database comparison. NMR signals? MS or MSMS patterns?

Response: Thank for the comment. Actually, we performed the LC-MS/MS of the EtOAc extract to GNPS platform (http://gnps.ucsd.edu) to establish the molecular clusters by molecular networking. Annotation of the nodes in clusters with the help of GNPS database and our laboratory database (LC-MS/MS spectra with more than 100 eremophilane-type sesquiterpenes) allowed the match of a number of known eremophilanes, which were previously isolated from other marine-derived fungi in our laboratory (Phytochemistry 2021, 192, e112978; Tetrahedron 2018, 74, 7310-7325; J. Nat. Prod. 2016, 79, 1035−1047). Additional annotation of the MS data in the cluster near that of eremophilanes (m/z 430-490) afforded a number of chlorine-bearing metabolites, which can’t hit in GNPS database and our database for terpenoids. The 1H NMR data of the EtOAc extract exhibited the methyl groups and olefinic protons are comparable to the relevant features of known eremophilanes. Since our in-house library (LC, ESIMS/MS, 1H and 13C NMR spectra of terpenoids) is immature to open for usage, we revised the sentences in text to add the protocol of LC-MS annotation by GNPS, and part of molecular clusters is provided in SI section (Figure S104). The description in the text is revised to” With the aim to continue our discovery of bioactive natural products from marine-associated organisms, a marine sponge (Xestospongia testudinaria)-associated fungus strain Penicillium copticola WZXY-m122-9 was selected for chemical examination. The LC-MS/MS data of the EtOAc extract of the cultured fungus were processed into a molecular network using MZmine and the GNPS platform (http://gnps.ucsd.edu), which allowed the formation of the spectral nodes into clusters. Annotation of the nodes in a cluster with m/z values of 200 to 300 by GNPS MS/MS spectral library matched eremophilanes including dehydropetasol (m/z 233 [M+H]+), dihydrosporogen AO-1 (m/z 251 [M+H]+), hydroxyphomenone (m/z 267 [M+H]+), sporogen AO-1 (m/z 248 [M+H]+) and penicilleremophilane X (m/z 279 [M+H]+). In addition, a cluster with the nodes ranging from m/z 430 to 480 which present chlorine feature ([M]+/[M+2]+ = 3:1) did not hit in database, suggesting a group of untapped metabolites.”

  • Some figures are blurry, please provide figures with higher resolution. (Figures 4, 7, 8)

Response: The resolutions of Figures 4, 7 and 8 are improved, and Figure 7 for ECD curves are combined for more visible.

  • Line 90, double check Figure 3, panel C.

Response: The mark in Figure 3 (panel C) is deleted.

Reviewer 2 Report

The manuscript entitled “Eremophilane-type sesquiterpenes from a marine-derived Penicillium copticola fungus with antitumor and neuroprotective activities” described the isolation and structural elucidation of twelve new compounds. This is a good work that could be accepted for publication after minor changes as described below.

-          Please change the color of 1 and 2 in Figure 1 to black.

-          Compounds 1, 2, and 10 are not novel compounds. A similar skeleton was reported previously by Kosemura et al. (Bull. Chem. Soc. Jpn. 65, 926, 1992).

-          Please provide the Figure containing key HMBC correlations of new compounds in Supporting information.

-          Please provide ECD spectra of compounds 9 and 10.

-          The NOE correlations of H-7 and H-5 could be found in both compounds 11 and 12. This indicated that the aglycone part of 11 and 12 are enantiomers based on the ECD data and the specific rotation as well. Please check and revise them. Alternatively, ECD data of the hydrolyzed products of 11 and 12 should be recorded to give more evidence.

-          Please move the position of Scheme 1 to the Discussion part. The discussion regarding the biosynthesis of new compounds in the Conclusion part should be moved to the Discussion part.

-          Please provide the complete conditions of semi-prep HPLC and the hydrolysis condition of 11 and 12 in the Material and methods part.

Others:

-          L61: “butane unit” is wrong, please revise

-          L64: “a methyl”, delete “substitution”

-          L104: OH-4’ and OH-5’: “br” to “brs”

-          L155: add [ref] for cpd JBIR-27

-          L200 and L231: Ref [15], not [16]

-          L213: Glc is Glucose unit, not Glu

-          L374: petroleum ether?

-          L377: add the reference of the Mosher method

-          L382: add the reference of the method

-          Please revise the chemical structures of 1 and 2 in Supporting information file.

Author Response

Answer to reviewer-2

  • Please change the color of 1 and 2 in Figure 1 to black.

Response: I did it accordingly.

  • Compounds 1, 2, and 10 are not novel compounds. A similar skeleton was reported previously by Kosemura et al. (Bull. Chem. Soc. Jpn. 65, 926, 1992).

Response: Thank for the comment. I agree that compounds 1, 2, and 10 are not novel compounds due to one analog reported in literature. In conclusion section, I revised it to “which are uncommonly found in nature” since this is the second report of the relevant skeleton in this work.

  • Please provide the Figure containing key HMBC correlations of new compounds in Supporting information.

Response: The key HMBC and COSY correlations of 1-10 are provided in SI section (Figure S105),

  • Please provide ECD spectra of compounds 9 and 10.

Response: The experimental and calculated ECD spectra of 9 and ICD spectra of 10 are provided in Figure 3.

  • The NOE correlations of H-7 and H-5 could be found in both compounds 11 and 12. This indicated that the aglycone part of 11 and 12 are enantiomers based on the ECD data and the specific rotation as well. Please check and revise them. Alternatively, ECD data of the hydrolyzed products of 11 and 12 should be recorded to give more evidence.

Response: Thank you for the valuable comment. Indeed, the experimental ECD curves and specific rotations of both 11 and 12 are in opposite signs, leading to the assumption that both compounds with enantiomer form. However, the 1H NMR data including the proton spins are somehow difference, suggesting bother compounds are stereoisomers rather than enantiomers. I carefully rechecked the NOE data of 11 and 12, and found the NOE interaction between H-5 and H-8 in 11 but the NOE interaction between H-5 and H-7 in 12. These findings suggested both 11 and 12 maintaining the configuration of C-5 with difference for remaining stereogenic centers. These assignments are in agreements with penostatins I and F (nucleus part) (Org. Lett. 2012, 14, 4738-4741). The experimental ECD curves of the hydrolyzed products of 11 (11a) and 12 (12a) are provided in Figure 7 for comparison.

  • Please move the position of Scheme 1 to the Discussion part. The discussion regarding the biosynthesis of new compounds in the Conclusion part should be moved to the Discussion part.

Response: Thank you for the suggestion. I did the revision to move the biogenetic depiction and Scheme 1 to the Discussion section.

  • Please provide the complete conditions of semi-prep HPLC and the hydrolysis condition of 11 and 12 in the Material and methods part.

Response: The detail procedure and conditions for semi-preparative HPLC separation are provided in the experimental section, and the Rt for each compound is added. The hydrolysis procedure of 11 and 12 is also provided in the experimental section.

  • L61: “butane unit” is wrong, please revise.

Response: I revised the term ““butane unit” to “a segment in ring A”.

  • L64: “a methyl”, delete “substitution”

Response: I revised “methyl substitution” to “methyl group.

  • L104: OH-4’ and OH-5’: “br” to “brs”

Response: Thank for the suggestions, I revised them accordingly, and the same errors in Table 2 are also corrected.

  • L155: add [ref] for cpd JBIR-27

Response: Reference [9] is cited after JBIR-27.

  • L200 and L231: Ref [15], not [16]

Response: I revised them accordingly.

  • L213: Glc is Glucose unit, not Glu

Response: In text, I revised it accordingly.

  • L374: petroleum ether?

Response: In experimental section, I corrected “Petroether” to “petroleum ether”.

(15) L377: add the reference of the Mosher method

Response: A reference “[27] Kusumi, T.; Hamada, T.; Ishitsuka, M.O.; Ohtani, I.; Kakisawa, H. Elucidation of the relative and absolute stereochemistry of lobatriene, a marine diterpene, by a modified Mosher method. J. Org. Chem. 1992, 57, 1033-1035.” is cited for the Mosher method.

(16) L382: add the reference of the method

Response: A reference “[28] Di Bari, L.; Pescitelli, G.; Pratelli, C.; Pini, D.; Salvadori, P. Determination of absolute configuration of acyclic 1,2-diols with Mo2(OAc)4. 1. Snatzke's method revisited. J. Org. Chem. 2001, 66, 4819-4825.” is cited for the Snatzke's method.

(17) Please revise the chemical structures of 1 and 2 in Supporting information file.

Response: The structures of 1 and 2 in SI section are corrected.

Reviewer 3 Report

A detailed chemical examination of the marine sponge-derived fungus  Penicillium copticola led to afford 12 undescribed natural products, of which ten were eremophilane-type sesquiterpenes and two were glucosides. Eremophilanes 1, 2 and 10 were characteristic of a phenyl moiety in ring B companying a methyl migration from C-5 to C-9, which were new scaffold of eremophilanes. The incorporation of a chlorinated phenylacetic unit in 39 was rarely found in nature, which represented the second isolation of relevant analogs from fungi. Analogs 11 and 12 were characteristic of an unprecedented PKS scaffold bearing a glucose unit. Moreover, analog 8 exhibited selective inhibition against human non-small cell lung cancer cells (A549), whereas analog 7 showed neuroprotective effect. These intriguing and important findings enriched the chemical diversity of eremophilanes and extended their bioactivities to neuroprotection, which will capture extensive attention from researchers.

This manuscript was well-written, only minor revisions before it is considered for publication.

1. P1L38: Please revise the sentence as ‘ ...fungal genera of Acremonium [6], Penicillium [7-10], Cochliobolus [11], Phomopsis [12], and Cryptosphaeria [13].’

2. P1L43: ‘sesquiterene-related    sesquiterpene-related

3. P3L104: ‘OH-4 (δH 9.01, br) and OH-5 (δH 9.76, br)    OH-4 (δH 9.01, brs) and OH-5 (δH 9.76, brs)

4. It is better to replace the ‘3’ and ‘enanti-3’ by ‘(3R,4R,5R,6R,7R)-3’ and ‘(3S,4S,5S,6S,7S)-3’ respectively in Figure 3C. Please do the similar revisions for other figures.

5. P5L131: The NMR data ‘H-3′ (δH 6.88, d, J = 8.2 Hz), H-2′ (δH 6.97, dd, J = 1.5, 8.2 Hz), and H-6′ (δH 7.18, d, J = 1.5 Hz)’ were not consistent with those recorded in Table 2. Please check it.

6. P5L134: According to the data in Table 3, please revise the sentence as ‘...the correlations of H-2′ to the nonprotonated carbons C-3′ (δC 119.8) and C-4′ (δC 152.5)’.

7. The chemical structure of compound 7 in Figure 1 was wrong while its structure in Figure S52S58 was correct. And please revise the reference for JBIR-27 as ‘[14]’ (P5L149).

8. As C-7 of compound 9 was a nonprotonated carbon, please check the sentence ‘...the COSY couplings of H-8 (δH 3.60, brd, J = 4.5 172 Hz) with H-7 and H-9 (δH 5.40, brd, J = 4.5 Hz)...’ (P6L172). And the subscript format for ‘H2-6 (δH 1.68, 1.69)’ (P6L174).

9. P7L198 & P8L207: Superscript format for ‘sp3’.

10. Please revise the reference for penostatin F as ‘[15]’ (P8L200).

11. As described on P11, the known compounds phomenone, JBIR28, and JBIR27 had been obtained, but these were not mentioned on P9 paragraph 2. And please add the references for these known ones.

12. The references [27-28] were cited on P10L274, but these two references were missing in the list of references.

13. Please correct the compound numberings 76 and 79 (P11L328). And the subscript format for ‘C18 ODS’ (P11L331).

14. Please add the signs ‘+’ or ‘–’ for the specific rotation values of the new compounds on P11 & P12.

15. P12L361 & L364: According to the data in Figure S59 & S67, please revise ‘C23H28ClO6’ as ‘C23H26ClO6’.

16. Please add the physical and chemical data for new compounds 11 and 12 on P12.

17. P12L372: ‘CH3Cl’    ‘CHCl3

Author Response

Answer to Reviewer-3

  • P1L38: Please revise the sentence as ‘ ...fungal genera of Acremonium [6], Penicillium [7-10], Cochliobolus [11], Phomopsis [12], and Cryptosphaeria [13].’

Response: Thank for the comments, I revised them accordingly.

  • P1L43: ‘sesquiterene-related’ →  ‘sesquiterpene-related’

Response: The sentences are reorganized and the typos are corrected accordingly.

  • P3L104: ‘OH-4′ (δH01, br) and OH-5′ (δH 9.76, br)’ →  ‘OH-4′ (δH 9.01, brs) and OH-5′ (δH 9.76, brs)’

Response: The errors are corrected accordingly, and these in Tables are also corrected.

  • It is better to replace the ‘3’ and ‘enanti-3’ by ‘(3R,4R,5R,6R,7R)-3’ and ‘(3S,4S,5S,6S,7S)-3’ respectively in Figure 3C. Please do the similar revisions for other figures.

Response: In the revised version, Figure 3 has been revised to add the configurations of model molecules for ECD calculation according to the referee’s suggestion.

  • P5L131: The NMR data ‘H-3′ (δH88, d, J = 8.2 Hz), H-2′ (δH 6.97, dd, J = 1.5, 8.2 Hz), and H-6′ (δH 7.18, d, J = 1.5 Hz)’ were not consistent with those recorded in Table 2. Please check it.

Response: The wrong numbering 5’ in Table 2 is corrected to 3’.

  • P5L134: According to the data in Table 3, please revise the sentence as ‘...the correlations of H-2′ to the nonprotonated carbons C-3′ (δC8) and C-4′ (δC 152.5)’.

Response: I revised the wrong assignments in text accordingly.

  • The chemical structure of compound 7 in Figure 1 was wrong while its structure in Figure S52S58 was correct. And please revise the reference for JBIR-27 as ‘[14]’ (P5L149).

Response: Thank for the valuable comment, structure 7 in Figure 1 is corrected. Reference [14] is deleted as it is identical to Ref. [9].

  • As C-7 of compound 9 was a nonprotonated carbon, please check the sentence ‘...the COSY couplings of H-8 (δH60, brd, J = 4.5 172 Hz) with H-7 and H-9 (δH 5.40, brd, J = 4.5 Hz)...’ (P6L172). And the subscript format for ‘H2-6 (δH 1.68, 1.69)’ (P6L174).

Response: The sentence is rephrased to “the COSY coupling between H-8 (δH 3.60, brd, J = 4.5 Hz) and H-9 (δH 5.40, brd, J = 4.5 Hz)”, and the subscript of H2-6 is introduced.

  • P7L198 & P8L207: Superscript format for ‘sp3’.

Response: I corrected it accordingly.

  • Please revise the reference for penostatin F as ‘[15]’ (P8L200).

Response: Thank for the comment, the Ref. is corrected to [14] in rvised version.

  • As described on P11, the known compounds phomenone, JBIR28, and JBIR27 had been obtained, but these were not mentioned on P9 paragraph 2. And please add the references for these known ones.

Response: In the revised text, phomenone, JBIR28, and JBIR27 are added, and the references for the known compounds are cited.

  • The references [27-28] were cited on P10L274, but these two references were missing in the list of references.

Response: Ref. [27] and [28] are cited in the text, and are provided in the Reference section.

[27] Rani, R.; Kumar V. When will small molecule lactate dehydrogenase inhibitors realize their potential in the cancer clinic? Future Med. Chem. 2017, 11, 1113-1115.

[28] Shi, Y.; Pinto, B.M. Human lactate dehydrogenase a inhibitors: a molecular dynamics investigation. PLoS One. 2014, 9, e86365.

(13) Please correct the compound numberings 76 and 79 (P11L328). And the subscript format for ‘C18 ODS’ (P11L331).

Response: The wrong numberings of 76 and 79 in the experimental section are corrected to 11 and 12.

(14) Please add the signs ‘+’ or ‘–’ for the specific rotation values of the new compounds on P11 & P12.

Response: The signs of specific rotations are added in the experimental section, and the chemical and physical data of 11 and 12 are supplied.

  • P12L361 & L364: According to the data in Figure S59 & S67, please revise ‘C23H28ClO6’ as ‘C23H26ClO6’.

Response: Thank you. I revised it accordingly.

  • Please add the physical and chemical data for new compounds 11 and 12 on P12.

Response: The chemical and physical data of 11 and 12 are supplied.

  1. P12L372: ‘CH3Cl’ → ‘CHCl3

Response: Thank you. I corrected the typo.
